# Diagnostic Utility of Metalloproteinases from Collagenase Group (MMP-1, MMP-8 and MMP-13) in Biochemical Diagnosis of Ovarian Carcinoma

**DOI:** 10.3390/cancers16233969

**Published:** 2024-11-26

**Authors:** Aleksandra Kicman, Ewa Gacuta, Rafał Marecki, Michał Stanisław Kicman, Monika Kulesza, Ewa Klank-Sokołowska, Paweł Knapp, Marek Niczyporuk, Maciej Szmitkowski, Sławomir Ławicki

**Affiliations:** 1Department of Aesthetic Medicine, The Faculty of Pharmacy, Medical University of Białystok, 15-267 Białystok, Poland; niczy.ma@gmail.com; 2Department of Perinatology, University Clinical Hospital of Bialystok, 15-276 Białystok, Poland; sunnyeve@wp.pl; 3Department of Psychiatry, The Faculty of Medicine, Medical University of Białystok, 15-272 Białystok, Poland; marecki.rafal.96@gmail.com; 4Independent Researcher, 15-213 Białystok, Poland; michal1kicman@gmail.com; 5Department of Population Medicine and Lifestyle Diseases Prevention, The Faculty of Medicine, Medical University of Białystok, 15-269 Białystok, Poland; monika.kulesza@sd.umb.edu.pl; 6University Cancer Center, University Clinical Hospital of Bialystok, 15-276 Białystok, Poland; ewaklank@wp.pl (E.K.-S.); pawel.knapp@umb.edu.pl (P.K.); 7Department of Biochemical Diagnostics, The Faculty of Medicine, Medical University of Białystok, 15-269 Białystok, Poland; msz@umb.edu.pl

**Keywords:** MMP-1, MMP-8, MMP-13, collagenases, matrix metalloproteinases, markers, HE4, CA125, ROMA, ovarian carcinoma, plasma concentration

## Abstract

Ovarian carcinoma (OC) is one of the most common gynecological cancers. Due to its asymptomatic course and lack of screening tests, the disease is detected in advanced stages, which translates into an unfavorable prognosis for patients. Therefore, it is appropriate to seek diagnostic methods for OC, which include the determination of tumor markers. Enzymes of the matrix metalloproteinase (MMP) group, which are involved in all stages of the process of carcinogenesis, have particular potential. Collagenases (MMP-1, MMP-8 and MMP-13) are a poorly studied group of enzymes from the 28 MMP group. Our study suggests that they may be postulated as new diagnostic markers of OC. However, the unequivocal confirmation of their potential requires further research.

## 1. Introduction

Ovarian carcinoma (OC) is one of the most common cancers of the reproductive tract. In 2022, 46,232 cases of OC were diagnosed in Europe and this number is steadily increasing. According to estimates by the World Ovarian Cancer Coalition (WOCC), in 2050, the number of cases will reach 75,570. In Asian or African countries, an even greater increase in incidence will be observed. This means that in the world population, in 2050, 8 million women will contract OC. In 2024 alone, 477,000 women will be diagnosed with OC with 70% of patients dying from the disease (October 2024 statistics) [1,2].

Unfortunately, OC, in addition to its increasing incidence, is characterized by an unfavorable prognosis. Around 60–70% of women are diagnosed with late-stage disease, most commonly stages III (tumor involves 1 or both ovaries or fallopian tubes, or peritoneal with cytologically or histologically confirmed spread to the peritoneum outside the pelvis and/or metastasis to the retroperitoneal lymph nodes) or IV (distant metastasis excluding peritoneal metastases), according to the classification of the International Federation of Gynecology and Obstetrics (FIGO) [2,3,4]. According to the prognosis, the 5-year survival rate decreases with increasing OC stage; for patients in stage III, it is 27%, while in stage IV, it is 13% [2,4]. OC, especially at later stages, has an unfavorable tendency to recur, with a median progression-free survival of 18 months [3].

The most important problems with OC diagnostics diagnosis are related to the mostly asymptomatic course of the disease and the lack of effective screening tests [2,5]. Currently, transvaginal ultrasonography (TVS) and the determination of classical tumor markers—CA125 and/or HE4, alone or in combination with consideration of the patient’s menopausal status in the Risk of Ovarian Malignancy Algorithm (ROMA)—are used in the diagnosis of OC. However, these methods have limited sensitivity and specificity [2,5,6]. However, while the histopathological result is still considered the “gold standard”, it requires surgery, with potential complications [5]. The introduction of a new method of early diagnosis of OC into clinical practice will allow the faster introduction of treatment and a significant improvement in prognosis—the 5-year survival rate for women in stage II is 70%, while in stage I, it is already 90% [2,6,7]. Among the methods postulated is the determination of concentrations of tumor markers in body fluids, such as peripheral blood [2,5,8,9].

Among the molecules that are most often considered as potential markers are matrix metalloproteinases (MMPs). MMPs are enzymes with proteolytic properties whose activity is associated with all stages of carcinogenesis [10,11,12]. The pathological activity of MMPs is closely related to the pathogenesis of OC [10,11,13,14] and preliminary studies indicate the potential of these enzymes as new markers, determined from peripheral blood in the diagnosis of this condition [10,15,16,17]. The present study is a continuation of scientific research on the usefulness of MMPs as novel markers in the diagnosis of OC. Collagenases are a group of MMPs; they include MMP-1, MMP-8 and MMP-13. Currently, these enzymes are very poorly studied and their potential as tumor markers is not understood. The aim of this study was to determine the diagnostic utility of collagenase group metalloproteinases—MMP-1, MMP-8 and MMP-13—in patients with OC, compared to women with benign ovarian lesions (*Serous cystadenomas*) and healthy subjects in comparison with classical markers (CA125 and HE4) and the ROMA.

## 2. Materials and Methods

### 2.1. Group Descriptions

Table 1 presents a description of the groups of patients qualified for the study, while Figure 1 shows the study flowchart. The experiment included 120 patients with *ovarian carcinoma* (OC). Each patient was classified into the appropriate stage according to the FIGO classification (I-V). On the basis of histopathological examinations performed on material collected intraoperatively, the histological diagnosis of OC was determined (types: *high-grade serous, endometrioid, mucinous and clear cell carcinoma*), and this evaluation was performed at the in-hospital diagnostic stage. Before any treatment was instituted, each patient had baseline blood tests, including the determination of CA125 and HE4 concentrations, ultrasound (US), X-ray, computed tomography (CT) or magnetic resonance imaging (MRI). Patients with known renal failure were excluded from the study, due to high HE4 concentrations. Before any treatment, written informed consent and medical history were obtained from all participants.

The control group included 70 patients with benign ovarian lesions (BL group), namely *Serous cystadenomas*, and 50 healthy women (HS group). BL patients were treated at the Department of Gynecology of the University Clinical Hospital in Bialystok from 2008 to 2012 and at the University Oncology Center of the University Clinical Hospital in Bialystok from 2019 to 2023. Confirmation of the benign nature of the malignant lesions in these patients was achieved on the basis of histopathological examination performed on material collected intraoperatively. Healthy female volunteers who were eligible for the study were recruited when they had their annual preventive examinations (laboratory tests, cervical cytology, abdominal ultrasound). All women in HS group were in post-menopausal status. Women with a positive clinical history of gynecological conditions were excluded from the study. The women underwent a gynecological check-up prior to the collection of material for the study. Written consent to participate in the study and their medical histories were obtained from all participants. 

In patients from the *ovarian carcinoma* group and benign ovarian lesion group were calculated using ROMA—in accordance with a previous paper published by our research team [2]. The prerequisite for determining the value of the ROMA was the presence of a masses within the ovary(s).

-For premenopausal women, a score of ≥11.4% was considered a high risk of OC, while <11.4% was considered a low risk.-For postmenopausal women, a score a value of ≥29.9% was considered high OC risk, while <29.9% was considered low risk.

The ROMA was not calculated in women who had undergone previous oncological treatment.

This study was performed in accordance with the Declaration of Helsinki. The protocol was authorized by the local Ethical Committee of the Medical University of Bialystok (committee approval number, APK.002.420.2021; approval date 21 October 2021). All patients gave informed consent to participate in the study.

### 2.2. Biochemical Assays

The test material was venous blood, collected from the elbow vein for anticoagulant lithium heparin. Within an hour of blood collection, the material was centrifuged at 1810× *g* for 10 min to obtain plasma. After centrifugation, the plasma was immediately separated from the morphotic mass and preserved at −81 °C until the day of biochemical determination. The tested collagenases were determined by enzyme-linked immunoassay (ELISA) with using kits from R&D Systems (Minneapolis, MN, USA)—MMP-1: Human Total MMP-1 DuoSet ELISA, cat. no. DY901B; MMP-8: Human Total MMP-8 DuoSet ELISA, cat. no. DY908; and MMP-13: Human Total MMP-13 DuoSet ELISA; cat. no. DY511. The assay procedure was performed according to the manufacturer’s recommendations included with each kit. Both standards and patient samples were assayed in duplicate. Plates were read by using a wavelength of 450 nm and a correction set at 540 nm by microplate spectrophotometer, BioTek EPOCH. The precision of the kits was set by the manufacturer—MMP-1: 10% (intra-assay) and 12% (inter-assay); MMP-8: 2.9% (intra-assay) and 7.1% (inter-assay); and MMP-13: 10% (intra-assay) and 12% (inter-assay). Routine markers HE4 and CA125 were measured by the chemiluminescent microparticle immunoassay (CMIA), in accordance with the manufacturer’s protocols on the Cobas e411 instrument using reagents from ROCHE Diagnostics (Rotkreuz, Switzerland, Roche Elecsys HE4 and Roche Elecsys CA125 II).

### 2.3. Statistical Analysis

Statistical analysis was performed using statistical PQStat Software (v.1.8.4.162, Poznań, Poland), while graphical processing was performed using GraphPad Prism 5 Software (GraphPad Software, La Jolla, CA, USA). The normality of distribution was assessed using the Shapiro–Wilk test. Further analyses were performed using non-parametric tests—the Kruskal–Wallis test with the Conover–Iman post hoc test for group comparisons and the Spearman rank correlation test to determine correlations between parameters. Using the ROC curve, an assessment of the diagnostic reliability and diagnostic power of the tests was performed with the determination of the optimal cutoff point for MMP-1 (74.145 ng/mL), MMP-8 (240.425 ng/mL), MMP-13 (128.914 ng/mL), HE4 (57.2 U/mL), CA125 (30.5 U/mL) and the ROMA (16.19%). The procedure of the experiment is shown in the flowchart (Figure 1). 

## 3. Results

### 3.1. Plasma Concentrations of MMP-1 in Patients with Ovarian Carcinoma (OC) and Benign Lesions (BLs) and in Healthy Subjects (HSs)

Higher levels of MMP-1 were found in OC patients (median: 263.0 ng/mL) compared to BL patients (median: 70.61 ng/mL; *p* < 0.000001) and HS patients (median: 9.928 ng/mL; *p* < 0.000001). This is shown in Figure 2 and in Table 2.

### 3.2. Plasma Concentrations of MMP-8 in Patients with Ovarian Carcinoma (OC), Patients with Benign Lesions (BLs), and Healthy Subjects (HSs) 

The highest MMP-8 concentrations are observed in the HS group (median: 306.4 ng/mL) compared to women with BLs (median: 262.3 ng/mL; *p* < 0.000001) and OC patients (median: 178.9 ng/mL; *p* < 0.000001). MMP-8 concentrations are shown in Figure 3 and in Table 2. 

### 3.3. Plasma Concentrations of MMP-13 in Patients with Ovarian Carcinoma (OC), Patients with Benign Lesions (BLs), and Healthy Subjects (HSs) 

Significantly higher levels of MMP-13 were found in OC patients (median: 291.83 ng/mL) compared to BL patients (median: 67.29 ng/mL; *p* < 0.000001) and the HS group (median: 11.97 ng/mL; *p* < 0.000001). This is shown in Figure 4 and in Table 2. 

### 3.4. Plasma Concentrations of HE4 and CA125 in Patients with Ovarian Carcinoma (OC), Patients with Benign Lesions (BLs), and Healthy Subjects (HSs)

The highest levels of both HE4 and CA125 are found in OC patients (HE4 median: 162.85 U/mL; CA125 median: 150.95 U/mL) compared to BL (HE4 median: 55.90 U/mL; *p* < 0.000001; CA125 median: 22.25; *p* < 0.000001) and HS (HE4 median: 36.25 U/mL; *p* < 0.000001; CA125 median: 16.49 U/mL; *p* < 0.000001). HE4 and CA125 concentrations in the three groups are presented in Figure 5 and in Table 2.

### 3.5. Percentages of the ROMA in Patients with Ovarian Carcinoma (OC), Patients with Benign Lesions (BLs), and Healthy Subjects (HSs)

Statistically higher ROMA percentages were obtained in OC patients (median: 76.41%) compared to the BL group (median: 13.58%, *p* < 0.000001). The percentages of the ROMA in the study groups are shown in Figure 6 and Table 2.

### 3.6. Evaluation of Correlation by Spearman’s Method

After performing Spearman’s analysis for the parameters studied, no statistically significant results were obtained. Data from Spearman’s analysis are presented in the Appendix A.

### 3.7. Diagnostic Criteria of MMP-1, MMP-8, MMP-13, HE4, CA125 and the ROMA

The diagnostic criteria—diagnostic sensitivity (SE), diagnostic specificity (SP), positive predictive value (PPV) and negative predictive value (NPV)—determined for the OC group are shown in Table 3.

For single analyses, the highest SE value was obtained for the ROMA (90.83%), and among tested collagenases, the highest values of this parameter were found for MMP-1 and MMP-8 (81.66% for both MMPs). These values were higher than the SE value obtained for CA125 (80%), but did not exceed the SE value for HE4 (85%). The lowest value of the studied parameter was obtained for MMP-13 (77.50%). Performing combined analyses increased the SE value of the studied parameters. The combined analysis of MMPs and MMPs with the ROMA was associated with an increase in SE, and the value of these parameters exceeded the value of SE obtained for the ROMA in every case except in the analysis of MMP-8 + MMP-13 (89.17%). The highest value for combined analyses was obtained for MMP-1 + ROMA (98.33%).

In single analyses, the highest SP value was found for CA125 (98%), and this value exceeded the SP value for MMP-1 (94%), MMP-8 (84%), MMP-13 (94%), HE4 (92%) and the ROMA (94%). Performing combined analyses was associated with an increase in SP values; however, a higher SP value than in CA125 was obtained only for MMP-1 + MMP-8 + MMP-13 analysis (99.15%). The SP values are shown in Table 3.

The highest PPV was again obtained for CA125 (98.96%) and for the ROMA (97.32%), and a slightly lower PPV was found for MMP-1 (97.02%). Comparable PPVs are found for MMP-13 (96.875%) and HE4 (96.22%). Combined analyses were associated with an increase in PPV for combining MMP-1 + MMP-13 (99.13%), MMP-1 + MMP-8 + MMP-13 (99.30%), MMP-1 + ROMA (99.16%) and MMP-13 + ROMA (99.15%). These data are shown in Table 3.

Single analysis showed the highest NPV for the ROMA (81.03%). Among tested collagenases, MMP-1 achieved the highest NPV (68.11%) (NPV for MMP-8: 65.625%; MMP-13: 63.51%); this was lower than for HE4 (71.875%) and higher than for CA125 (67.12%). Combined analyses were associated with an increase in NPV for all tested combinations. For the combined analyses of MMPs, the highest NPV was obtained for MMP-1 + MMP-8 + MMP-13 (89.29%), and analyses of MMPs with the ROMA showed the highest value for MMP-1 + ROMA (96.08%). NPV values are shown in Table 3. 

### 3.8. Evaluation the Diagnostic Power of Tests (ROC Function) 

The diagnostic power of the tests was determined by evaluating the area under the ROC curve (AUC). The analysis of the ROC curve determines whether the performance of the selected test makes it capable of distinguishing normal from abnormal results. From a diagnostic point of view, the ideal test makes it possible to distinguish a healthy person from a sick person, with a sensitivity value of 100% and specificity of 100%. In such a test, the line of the ROC function will completely coincide with the Y-axis, and the AUC value will be equal to 1. For tests that are not diagnostically useful, making it impossible to distinguish between sick and healthy people, the AUC value will be close to 0.5 (the limit of the diagnostic usefulness of the test). The characteristics of the ROC curve are shown in Table 4 and Figure 7, Figure 8 and Figure 9. 

In individual analyses, the highest AUC value was obtained for MMP-1 (AUC = 0.9625, *p* < 0.000001), a value that exceeded AUCs obtained for HE4 (AUC = 0.943, *p* < 0.000001), CA125 (AUC = 0.909, *p* < 0.000001) and the ROMA (AUC = 0.955, *p* < 0.000001). Lower AUC values, but which still exceeded the 0.5 value, were obtained for MMP-8 (AUC = 0.859, *p* < 0.000001) and MMP-13 (AUC = 0.917, *p* < 0.000001). The evaluation of diagnostic power based on area under the ROC curve (AUC) in single analyses is shown in Figure 7 and in Table 4.

After performing combined analyses of MMPs, the highest AUC value was obtained for the combination of MMP-1 + MMP-8 + MMP-13 (AUC = 0.988, *p* < 0.000001). This value was higher than the AUCs obtained for HE4, CA125 and the ROMA. The combinations of MMP-1 + MMP-8 (AUC = 0.974, *p* < 0.000001) and MMP-1 + MMP-13 (AUC = 0.984, *p* < 0.000001) also yielded higher AUC values than for the classical markers. The lowest AUC value is found for MMP-8 + MMP-13 (AUC = 0.952, *p* < 0.000001)—in this case, the value was higher only than HE4 and CA125. The evaluation of diagnostic power based on area under the ROC curve (AUC) in combined analyses of MMPs is shown in Figure 8 and in Table 4.

The combined analyses of MMPs with the ROMA showed a higher AUC value than for ROMA as a single marker and HE4 and CA125. The highest AUC value was obtained for the combination of MMP-1 + ROMA (AUC = 0.997, *p* < 0.000001). A similar but slightly lower value was obtained for the combinations of MMP-8 + ROMA (AUC = 0.9825, *p* < 0.000001) and MMP-13 + ROMA (AUC = 0.990, *p* < 0.000001), and these values were also higher than the AUCs obtained for the ROMA and the classical markers. The evaluation of diagnostic power based on area under the ROC curve (AUC) in the combined analyses of MMPs with ROMA is shown in Figure 9 and in Table 4.

## 4. Discussion

Ovarian carcinoma is characterized by a steadily increasing incidence and an extremely unfavorable prognosis. Due to its usual asymptomatic course and lack of screening, most patients are diagnosed at the latest stages of the disease (stages III or IV, according to the FIGO classification). The 5-year survival rate for patients in stage III is 27%, and in stage IV, it is only 13% [1,2,8]. Currently, OC diagnosis is based on transvaginal ultrasonography (TVS) and the determination of tumor markers—CA125 and/or HE4. These markers, despite exhibiting good values in terms of diagnostic parameters, have limited usefulness [2,18]. In patients with masses present in the pelvis, the ROMA can be utilized [2,19]. In addition, the final diagnosis of OC is based on the result of histopathological examination; however, obtaining the sample requires performing a surgical procedure which is associated with the risk of health complications for patients [2,5].

The detection of OC at an earlier stage allows for the faster introduction of treatment and significantly improves the prognosis of patients [2,3,4]. Therefore, it is expedient to introduce new diagnostic methods that allow the earlier detection of this condition. These include biochemical methods that determine concentrations of tumor markers in the peripheral blood. Therefore, the aim of the present study was to determine the diagnostic utility of collagenases (MMP-1, MMP-8 and MMP-13), a group of enzymes belonging to matrix metalloproteinases, in the biochemical diagnosis of ovarian carcinoma in comparison with patients with benign ovarian lesions (Serous cystadenomas) and healthy women, compared with the routinely used HE4 and CA125 markers and the ROMA. This work is a continuation of our team’s previous studies on the usefulness of MMPs in the diagnosis of ovarian cancer [2].

MMP-1 expression and activity are found in the physiological structures of the ovary [11,20,21]. Importantly, MMP-1 expression is also found in biopsy material obtained from patients with OC and women with benign lesions. MMP-1 expression was higher in OC samples compared to benign replacements [11,22]. The increased tissue expression of MMP-1 may translate into increased concentrations of this enzyme in the bloodstream, which seems to be confirmed by our study, in which we found statistically higher MMP-1 concentrations in OC patients (median: 263.0 ng/mL) compared to women with benign lesions (median: 70.61 ng/mL; *p* < 0.000001) and a group of healthy women (median: 9.928 ng/mL; *p* < 0.000001). We are now the first team to determine MMP-1 concentrations in OC patients, so our results will be partially applicable to other cancer types. Higher levels of MMP-1 are found in the serum of patients with pancreatic cancer (mean: 8.7 ± 7.5 ng/mL) compared to healthy subjects (mean: 6.7 ± 4.9 ng/mL; *p* < 0.0001) [23] and in the plasma of patients with lung cancer (median: 1.78 ng/mL) compared to healthy subjects (median: 0.99 ng/mL; *p* < 0.001) [24]. It should be noted that the concentrations of MMP-1 obtained in our study are significantly higher than in the experiments of Xu et al. [23] and Li et al. [24]. This is probably due to the different types of kits selected for MMP-1 assays and may also indicate a particularly important role for MMP-1 in the pathogenesis of OC from which high concentrations of this enzyme in patients with this condition would result.

Unlike MMP-1, MMP-8 expression has not been demonstrated in physiological structures of the ovary [25]. However, the presence of mRNA for this enzyme is found in OC biopsy samples, which may indicate a role for MMP-8 in OC pathogenesis [11,14]. The importance of MMP-8 in the pathogenesis of this condition was also suggested by Stenman et al. [26], who found higher MMP-8 activity in fluid collected from malignant ovarian cysts compared to fluid collected from benign lesions (*p* = 0.001) and suggested its role in the increased invasiveness of OC. In our study, we showed that OC patients had lower levels of MMP-8 (178.9 ng/mL) compared to BL patients (Median: 262.3 ng/mL; *p* < 0.000001) and healthy subjects (Median: 306.4 ng/mL; *p* < 0.000001). The results we obtained may indicate that MMP-8 is involved in the pathogenesis of OC; however, its activity is limited to the tumor lesion itself—the reason for which may be the lower concentrations of MMP-8 in the peripheral blood of OC patients than in BL and HS. It is unfortunate that this is only an assumption, as we are the first team to determine MMP-8 concentrations in OC patients; additionally, this enzyme is currently poorly studied within oncology. Few reports have indicated the possibility of using MMP-8 to predict survival in patients with colorectal or head and neck cancer [27,28,29].

MMP-13, like MMP-8, is not expressed in ovarian tissue. The mRNA for this enzyme only appears in samples from OC patients, which may also suggest a role for MMP-13 in OC pathogenesis [11,14]. The higher expression of MMP-13 in OC patients is associated with a poorer prognosis, as expressed by the overall survival value [14]. In contrast to MMP-8, the activity of this enzyme in fluid collected from malignant and non-malignant ovarian cysts is low and does not differ between groups [26]. A single study by Hantke et al. [30] also indicates the presence of MMP-13 in the peritoneal fluid of patients in advanced stages of OC; women with higher levels of this enzyme had a worse prognosis of the disease course. Our acquired studies showed significantly higher concentrations of MMP-13 in OC patients (median: 291.83 ng/mL) compared to the BL group (median: 67.29 ng/mL; *p* < 0.000001) and healthy subjects (median: 11.97 ng/mL; *p* < 0.000001). Again, we were the first research team to determine changes in MMP-13 in blood levels in OC patients. To our knowledge, MMP-13 as a potential marker has only been studied in esophageal cancer by the team of Wang et al. [31], who found higher serum levels of this enzyme in patients with this type of cancer (median: 328 pg/mL) compared to healthy subjects (median: 255 pg/mL). These results partially agree with the data we obtained.

The changes in the concentrations of the collagenases tested suggest their usefulness as new tumor markers, and thus, we determined their diagnostic utility by evaluating their sensitivity (SE), diagnostic specificity (SP), negative (NPV) and positive predictive value (PPV), and power of the test (ROC-AUC curve). These results were related to the diagnostic parameters of routine markers and the ROMA. All analyses were performed for individual parameters or combined parameters. Thus, we were able to preliminarily establish the diagnostic potential of these enzymes. Although MMP-1, MMP-8 and MMP-13 as individual parameters showed good diagnostic values, most of their values were lower than the diagnostic parameter values obtained for the ROMA. However, it should be emphasized that the prerequisite for performing the ROMA is the presence of pelvic lesions. Therefore, the potential of the collagenases we studied is bidirectional.

First, MMP-1 and MMP-13 show preliminary potential as diagnostic markers of OC—in our study, we showed that the concentrations of these enzymes are significantly higher in patients with OC compared to the BL and HS groups. Importantly, MMP-1 (SE: 81.66%; SP: 94%; PPV: 97.02%, NPV: 68.11%; AUC: 0.9625) and MMP-13 (SE: 77.50%; SP: 94%; PPV: 96.875%, NPV: 63.51%, AUC: 0.917) showed similar or higher diagnostic values to CA125 (SE: 80%; SP: 98%; PPV: 98.96%, NPV: 67.12%; AUC: 0.909) and HE4 (SE: 85%; SP: 92%; PPV: 96.22%, NPV: 71.875%; AUC: 0.943). HE4 and CA125 are currently the routine markers for the biochemical diagnosis of OC, and despite their good diagnostic values, their concentrations are influenced by a number of physiological and pathological phenomena in a woman’s body; in addition, they have limited usefulness in the diagnosis of this disease [2,5,11,32]. Performing combined analyses of individual MMPs was associated with a further increase in the values of diagnostic parameters, with the greatest increase found in the case of combined analysis of MMP-1 + MMP-13 (SE: 95%; SP: 98%; PPV: 99.13%, NPV: 89.09%; AUC: 0.984). All this indicates the high preliminary potential of MMP-1 and MMP-13 as potential OC markers, especially when using the MMP-1 + MMP-13 combined assay.

The second direction of our research was to determine the usefulness of MMPs as auxiliary differentiation markers in combination with the ROMA. To achieve this, we performed combined analyses of individual MMPs with the ROMA. Although the ROMA as a stand-alone parameter shows very good values of diagnostic parameters (SE: 90.83%; SP: 94%; PPV: 97.32%, NPV: 81.03%; AUC: 0.955), performing combined analyses of individual MMPs with the ROMA was associated with a further increase in all diagnostic parameters. The highest increase is found for MMP-1 + ROMA analysis (SE: 98.33%; SP: 98%; PPV: 99.16%, NPV: 96.08%; AUC: 0.997). This indicates the potential of collagenases as auxiliary differentiation markers.

In conclusion, collagenases MMP-1 and MMP-13 show bidirectional potential in the biochemical diagnosis of ovarian cancer. Firstly, as diagnostic markers, they can act as stand-alone markers or in combination MMP-1 + MMP-13. Secondly, as auxiliary differentiation markers used in conjunction with ROMA, the combination of MMP-1 + ROMA has particularly high potential. However, these are preliminary studies, and unequivocally confirming the potential of collagenases as new markers in biochemical diagnosis of OC requires further research, especially with a larger number of patients.

Unfortunately, our work has a number of limitations. The first is the number of patients selected for the study. The number of patients who participated in the study (OC: 120; BL: 70; HS: 50) allowed us to obtain only preliminary studies. In future studies, the study and control groups will be increased to 1000. This will allow us to better assess the usefulness of the studied collagenases. In addition, we used only two analytical methods in our study—ELISA and CMIA. In future studies, we want to confirm the results obtained by these methods with analytical techniques as well, for example, using Luminex multiplex assay. The last limitation is that simultaneously with the determination of collagenases in peripheral blood, we did not perform determinations of their expression in tissue material. We plan to perform such determinations in the future, with a larger number of patients. Nevertheless, we feel that our manuscript is innovative and sets new directions for research in the biochemical diagnosis of *ovarian carcinoma*.

## 5. Conclusions

MMP-1 and MMP-13 have shown preliminary potential in the biochemical diagnosis of ovarian carcinoma—as diagnostic markers (in stand-alone or combined assays) and as auxiliary differentiation markers assayed together with the ROMA.

## Figures and Tables

**Figure 1 cancers-16-03969-f001:**
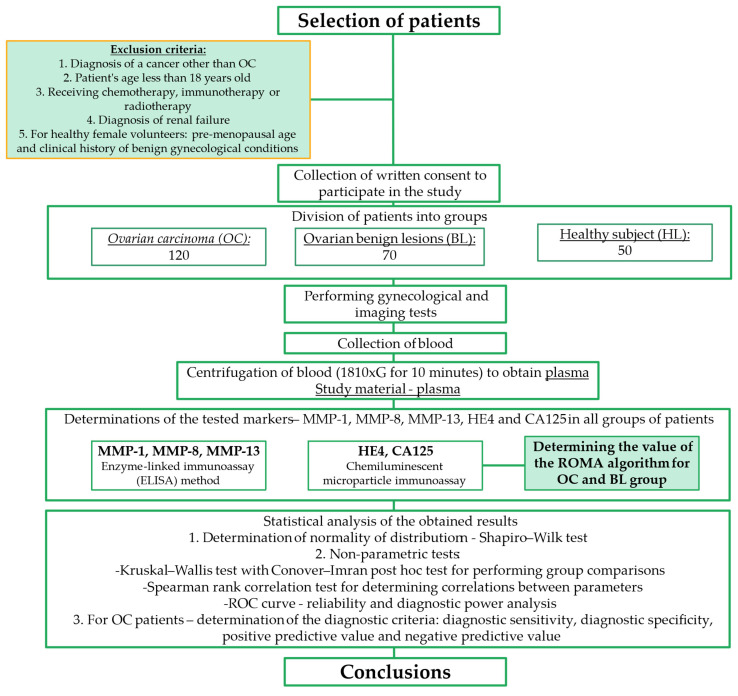
Flowchart of the experimental design.

**Figure 2 cancers-16-03969-f002:**
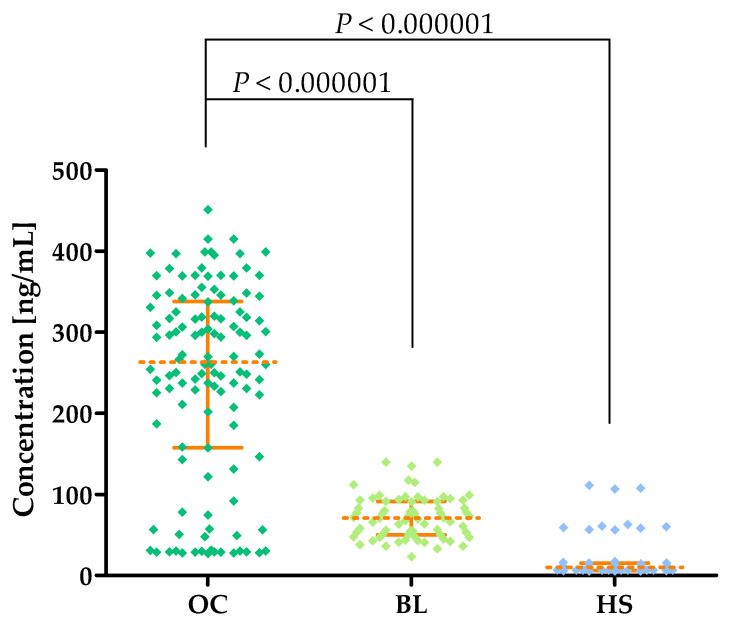
MMP-1 plasma concentrations (with marked median and interquartile ranges) in all tested groups: patients with ovarian carcinoma (OC) and benign lesions (BLs) and healthy subjects (HSs).

**Figure 3 cancers-16-03969-f003:**
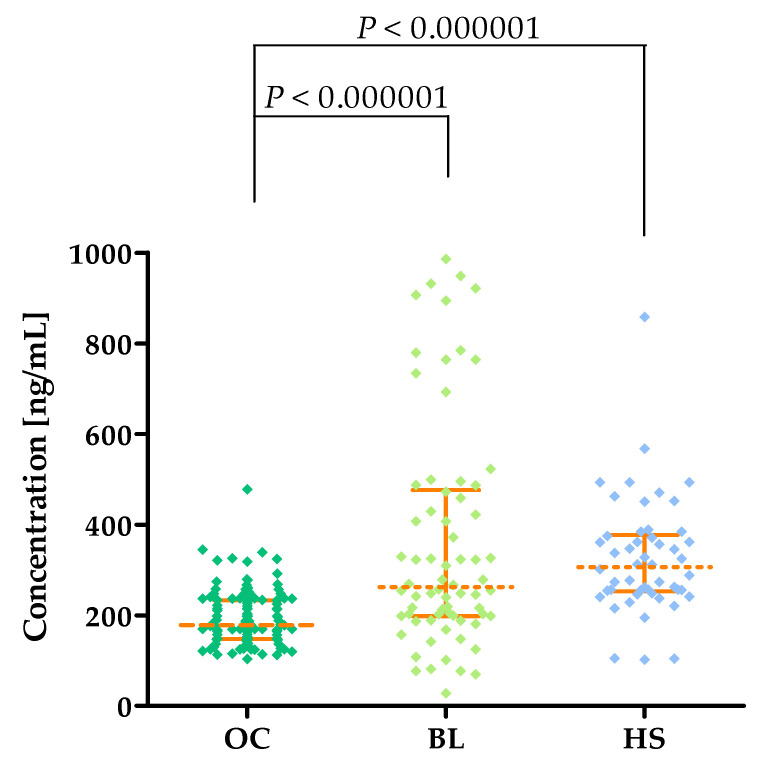
MMP-8 plasma concentrations (with marked median and interquartile ranges) in all tested groups: patients with ovarian carcinoma (OC) and benign lesions (BLs) and healthy subjects (HSs).

**Figure 4 cancers-16-03969-f004:**
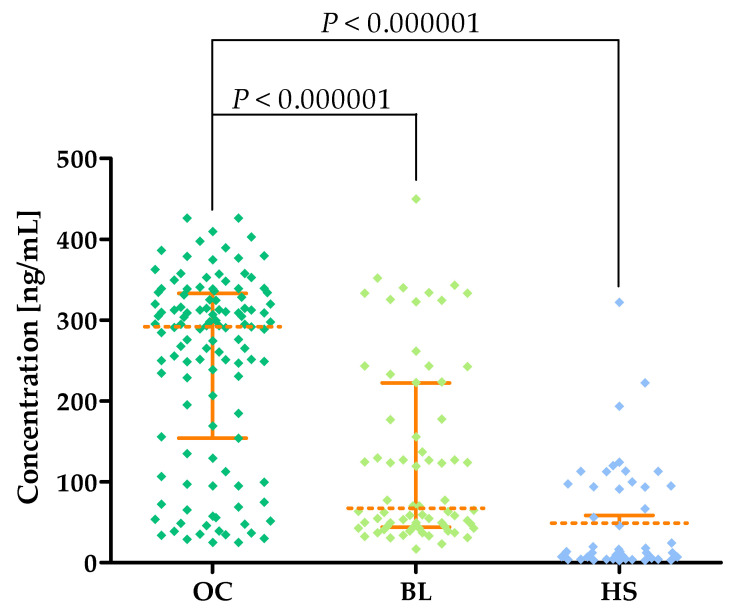
MMP-13 plasma concentrations (with marked median and interquartile ranges) in all tested groups: patients with ovarian carcinoma (OC), patients with benign lesions (BLs) and healthy subjects (HSs).

**Figure 5 cancers-16-03969-f005:**
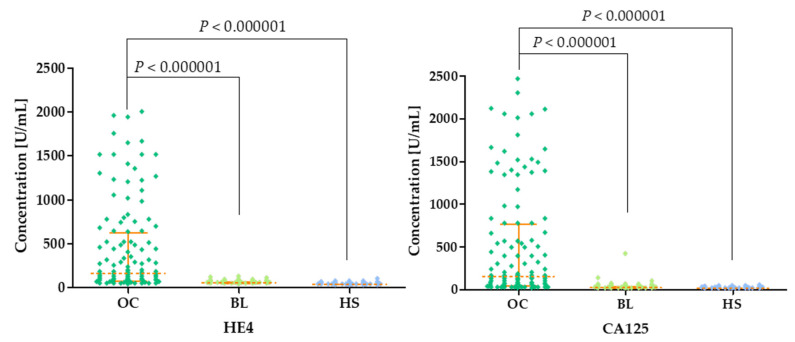
HE4 and CA125 plasma concentrations (with marked median and interquartile ranges) in all tested groups: patients with ovarian carcinoma (OC), patients with benign lesions (BLs) and healthy subjects (HSs).

**Figure 6 cancers-16-03969-f006:**
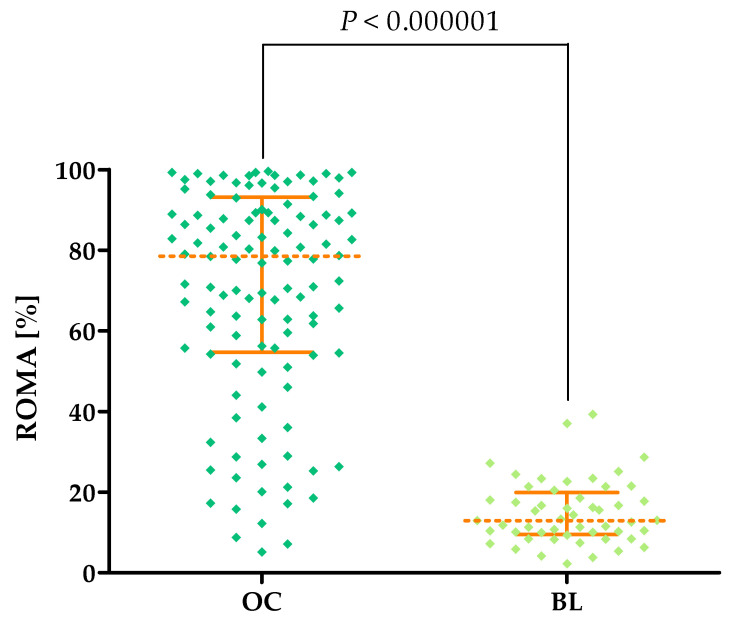
Percentages of the ROMA (with marked median and interquartile ranges) in all tested groups: patients with ovarian carcinoma (OC) and benign lesions (BLs).

**Figure 7 cancers-16-03969-f007:**
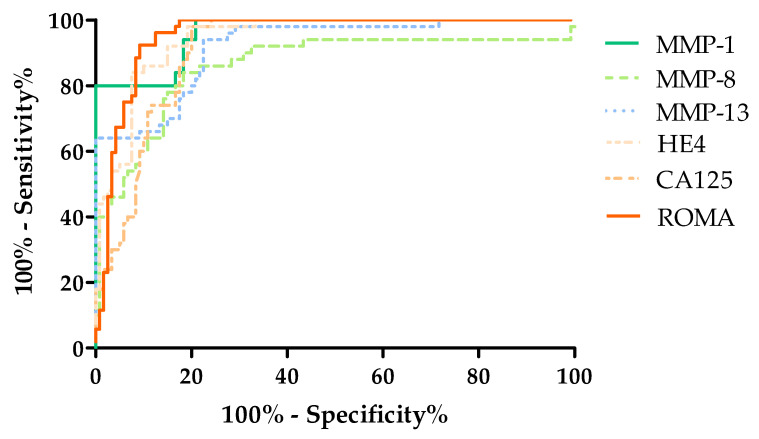
Evaluation of diagnostic power based on area under the ROC curve (AUC).

**Figure 8 cancers-16-03969-f008:**
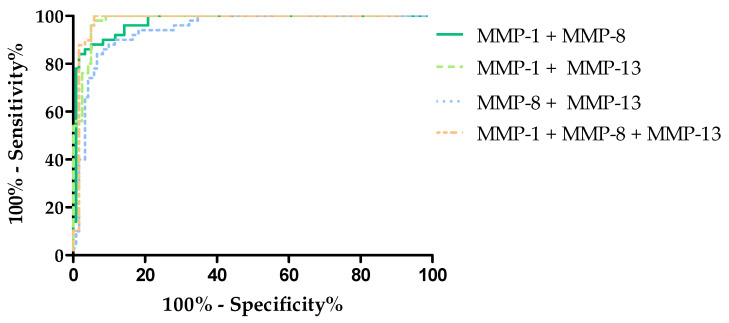
Evaluation of diagnostic power based on area under the ROC curve (AUC) for combined analyses of MMPs.

**Figure 9 cancers-16-03969-f009:**
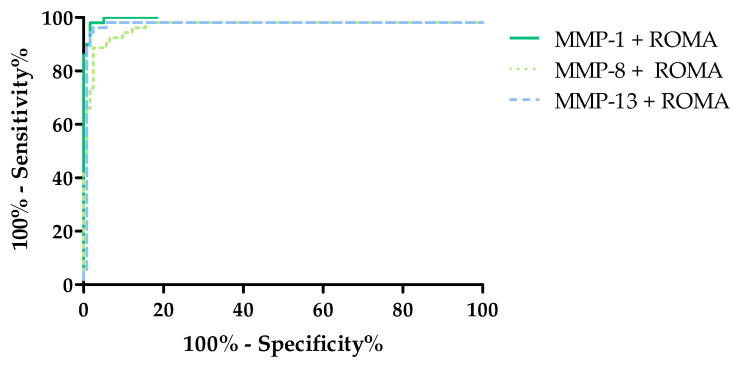
Evaluation of diagnostic power based on area under the ROC curve (AUC) for combined analyses of MMPs with ROMA.

**Table 1 cancers-16-03969-t001:** Characteristic of groups: ovarian carcinoma (OC) (histological diagnosis and stage, menopausal status and age), benign lesions (BLs) (histological diagnosis, menopausal status and age) and healthy subjects (HSs) (menopausal status and age).

Study Group—*Ovarian Carcinoma* (OC)
**Number of patients**
120 (100%)
**Median of age (range)**
60 (22–81)
**Histopathological diagnosis**
*High-Grade Serous Carcinoma*—82 (68%)
*Endometrioid Carcinoma*—32 (27%)
*Clear Cell Carcinoma*—1 (1%)
*Mucinous Carcinoma*—5 (4%)
**Tumor stage according to the International Federation of Gynecology and Obstetrics**
Stage I—19 (16%)
IA—3
IB—8
IC—8
Stage II—26 (22%)
IIA—11
IIB—6
IIC—9
Stage III—29 (24%)
IIIA—18
IIIB—7
IIIC—4
Stage IV—46 (38%)
**Menopausal status**
Pre-menopause—22 (18%)
Post-menopause—98 (82%)
**Control group—benign ovarian lesion (BL)**
**Number of patients**
70 (100%)
**Median age (range)**
48.5 (16–80)
**Histopathological diagnosis**
*Serous cystadenomas*—70 (100%)
**Menopausal status**
Pre-menopause—28 (40%)
Post-menopause—42 (60%)
**Control group—healthy subjects (HS)**
**Number of patients**
50 (100%)
**Median age (range)**
55 (46–77)
**Menopausal status**
Post-menopause—50 (100%)

**Table 2 cancers-16-03969-t002:** Plasma concentrations of MMP-1, MMP-8, MMP-13, HE4 and CA125; comparative markers CA125 and HE4; and value of the ROMA expressed in percentages (median, min–max range and interquartile range [IQR] in ovarian carcinoma (OC) patients, benign lesion (BL) patients and healthy subjects (HSs).

		Ovarian Carcinoma (*n* = 120)	Benign Lesions (*n* = 70)	Healthy Subjects (*n* = 50)
**MMP–1 [ng/mL]**	Median:	263.0	70.61	9.928
	Min–max range:	26.37–451.2	27.57–139.5	5.110–110.8
	IQR:	157.3–338.1	49.90–91.34	6.199–15.19
**MMP–8 [ng/mL]**	Median:	178.9	262.3	306.4
	Min–max range:	103.6–477.8	27.68–986.0	102.4–857.9
	IQR:	147.4–233.2	198.5–476.1	252.8–377.5
**MMP–13 [ng/mL]**	Median:	291.83	67.29	11.97
	Min–max range:	24.91–549.5	16.52–529.5	2.050–321.7
	IQR:	154.9015–331.64	44.381–211.1135	6.356–93.6175
**HE4** **[U/mL]**	Median:	162.85	55.90	36.25
	Min–max range:	28.93–1985	28.20–111.7	17.65–89.37
	IQR:	70.85–625.13	46.575–63.7	29.855–41.8975
**CA125 [U/mL]**	Median:	150.95	22.25	16.49
	Min–max range:	9.800–2742.00	5.800–410.3	1.110–39.94
	IQR:	46.34–766.00	14.75–37.40	11.2–20.87
**ROMA** **[%]**	Median:	76.41	13.58	
	Min–max range:	5.00–98.00	2.00–39.00	
	IQR:	36.61–95.13	9.09–18.80	

**Table 3 cancers-16-03969-t003:** Diagnostic parameter values—diagnostic sensitivity (SE), diagnostic specificity (SP), positive predictive value (PPV) and negative predictive value (NPV)—for the tested markers in single and combined analyses.

	SE [%]	SP [%]	PPV [%]	NPV [%]
**MMP-1**	81.66	94	97.02	68.11
**MMP-8**	81.66	84	92.45	65.625
**MMP-13**	77.50	94	96.875	63.51
**HE4**	85	92	96.22	71.875
**CA125**	80	98	98.96	67.12
**ROMA**	90.83	94	97.32	81.03
**Combined analyses of MMPs**				
**MMP-1 + MMP-8**	95.00	88.00	95	88.00
**MMP-1 + MMP-13**	95.00	98.00	99.13	89.09
**MMP-8 + MMP-13**	89.17	90.00	95.54	77.59
**MMP-1 + MMP-8 + MMP- 13**	95.00	99.15	99.30	89.29
**Combined analyses—MMPs with the ROMA**				
**MMP-1 + ROMA**	98.33	98.00	99.16	96.08
**MMP-8 + ROMA**	93.33	94.00	97.39	85.45
**MMP-13 + ROMA**	97.50	98.00	99.15	94.23

**Table 4 cancers-16-03969-t004:** Characteristics of ROC curve for parameters tested individually and in combination in the OC group (Red indicates statistical significance).

	AUC	SE_AUC_	95% CI	p (AUC = 0.5)
**MMP-1**	0.9625	0.0128	0.937–0.987	<0.000001
**MMP-8**	0.859	0.0367	0.787–0.931	<0.000001
**MMP-13**	0.917	0.0224	0.873–0.961	<0.000001
**HE4**	0.943	0.0161	0.912–0.975	<0.000001
**CA125**	0.909	0.0214	0.867–0.951	<0.000001
**ROMA**	0.955	0.0153	0.925–0.985	<0.000001
**Combined analyses of MMPs**				
**MMP-1 + MMP-8**	0.974	0.0106	0.953–0.994	<0.000001
**MMP-1 + MMP-13**	0.984	0.0073	0.969–0.998	<0.000001
**MMP-8 + MMP-13**	0.952	0.0155	0.921–0.982	<0.000001
**MMP-1 + MMP-8 + MMP- 13**	0.988	0.0077	0.973–1	<0.000001
**Combined analyses—MMPs with ROMA**				
**MMP-1 + ROMA**	0.997	0.00187	0.993–1	<0.000001
**MMP-8 + ROMA**	0.9825	0.00768	0.967–0.997	<0.000001
**MMP-13 + ROMA**	0.990	0.00811	0.974–1	<0.000001

## Data Availability

Research data can be provided upon request after consultation with the correspondent author (olakicman@gmail.com; slawicki@umb.edu.pl).

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
