# Peer review of "Diagnostic Utility of Metalloproteinases from Collagenase Group (MMP-1, MMP-8 and MMP-13) in Biochemical Diagnosis of Ovarian Carcinoma"

_cancers, 2024, doi:10.3390/cancers16233969_

Round 1
Reviewer 1 Report
Comments and Suggestions for Authors
The author evaluated the diagnostic utility of collagenases (MMP-1, MMP-8 and MMP-13) in the diagnosis of epithelial ovarian cancer (EOC) compared to HE4 and CA125 and 34 the ROMA. The authors concluded that MMP-1 and MMP-13 have shown preliminary potential as diagnostic markers and auxiliary markers to ROMA in biochemical diagnosis of EOC. This research cannot be used immediately in clinical practice. However, I think it will be interesting for cancers readers.
Major comments;
1. Did you think that the distribution of stage and histological subtype for EOC in your study population reflected the real-world situation in the Polish population?
2. All patients with benign ovarian lesion in your study had serous cystadenoma. In clinical situation, that is impossible. In previous studies for evaluating the diagnostic utility of candidate tumor makers in the diagnosis of EOC, various types of benign ovarian lesion were included.
Author Response
The author evaluated the diagnostic utility of collagenases (MMP-1, MMP-8 and MMP-13) in the diagnosis of epithelial ovarian cancer (EOC) compared to HE4 and CA125 and 34 the ROMA. The authors concluded that MMP-1 and MMP-13 have shown preliminary potential as diagnostic markers and auxiliary markers to ROMA in biochemical diagnosis of EOC. This research cannot be used immediately in clinical practice. However, I think it will be interesting for cancers readers.
Dear Reviewer.
We would like to thank You very much for your thorough and honest review of our manuscript “Diagnostic utility of metalloproteinases from collagenase group (MMP-1, MMP-8 and MMP-13) in biochemical diagnosis of ovarian carcinoma.” We will try to carefully answer all the Reviewer's questions and objections. Our answers will be written in green italics.
Major comments;
- Did you think that the distribution of stage and histological subtype for EOC in your study population reflected the real-world situation in the Polish population?
Thank You for Your comment. Currently, we do not have statistical data that show the distribution of the severity and histological subtype of EOC in Poland. Epidemiological data only provide insight into the number of incidences and deaths from EOC by year in Poland. However, samples for our study were collected from 2008-2012 and 2019-2023, which allows at least partial insight into the actual epidemiological situation. In addition, other scientific work conducted by our research team, however, on a different group of samples, collected in a different time frame, reveals a similar trend to ours. For example:
- Będkowska, G.E.; Gacuta, E.; Zajkowska, M.; Głażewska, E.K.; Osada, J.; Szmitkowski, M.; Chrostek, L.; Dąbrowska, M.; Ławicki, S. Plasma Levels of MMP-7 and TIMP-1 in Laboratory Diagnostics and Differentiation of Selected Histological Types of Epithelial Ovarian Cancers. J Ovarian Res 2017, 10, 39, doi:10.1186/s13048-017-0338-z. – In the work of Będkowska et al. we have similar proportions in the number of patients in each FIGO stage and histological subtype.
- Lawicki S, Będkowska GE, Gacuta-Szumarska E, Szmitkowski M. The plasma concentration of VEGF, HE4 and CA125 as a new biomarkers panel in different stages and sub-types of epithelial ovarian tumors. J Ovarian Res. 2013 Jul 2;6(1):45. doi: 10.1186/1757-2215-6-45. PMID: 23819707; PMCID: PMC3706238. – In the work of Lawicki et al. the statistics of the number of patients in both the individual FIGO stage and histological subtype are very similar to our work.
In addition, the quantitative proportions of patients in each FIGO stage and EOC histologic subtype are consistent with literature data relating to the global population. We presented these information in our review paper (Kicman A, Niczyporuk M, Kulesza M, Motyka J, Ławicki S. Utility of Matrix Metalloproteinases in the Diagnosis, Monitoring and Prognosis of Ovarian Cancer Patients. Cancer Manag Res. 2022 Nov 30;14:3359-3382. doi: 10.2147/CMAR.S385658. PMID: 36474934; PMCID: PMC9719685.) which is the first in a series of publications on the role of matrix metalloproteinases in the diagnosis of ovarian cancer.
- All patients with benign ovarian lesion in your study had serous cystadenoma. In clinical situation, that is impossible. In previous studies for evaluating the diagnostic utility of candidate tumor makers in the diagnosis of EOC, various types of benign ovarian lesion were included.
Thank You for Your comment, we are aware that all patients with benign lesions had serous cystadenomas and that in clinical practice this situation would not occur. This is due to several reasons. First, we wanted to maintain continuity with our previous publication (Kicman A, Gacuta E, Kulesza M, Będkowska EG, Marecki R, Klank-Sokołowska E, Knapp P, Niczyporuk M, Ławicki S. Diagnostic Utility of Selected Matrix Metalloproteinases (MMP-2, MMP-3, MMP-11, MMP-26), HE4, CA125 and ROMA Algorithm in Diagnosis of Ovarian Cancer. Int J Mol Sci. 2024 Jun 6;25(11):6265. doi: 10.3390/ijms25116265. PMID: 38892452; PMCID: PMC11173327.) in which patients with benign lesions were diagnosed exclusively with serous cystadenoma. Secondly, we classified a very small number of patients with other benign lesions of the ovary for the study. In this situation we decided to include only patients with serous cystadenomas thus obtaining a homogeneous group. However, we feel that a homogeneous group of patients with benign lesions does not affect the quality of our manuscript.
In conclusion, we sincerely thank You for such a positive evaluation of our manuscript. We hope that the answers we presented to the reviewer's questions will be satisfactory and our manuscript will be accepted for publication in the “Cancers”.
Best regards,
Aleksandra Kicman
prof. dr hab. Sławomir Ławicki
also, on behalf of all authors

Reviewer 2 Report
Comments and Suggestions for Authors
The manuscript titled ‘Diagnostic utility of metalloproteinases from collagenase group (MMP-1, MMP-8 and MMP-13) in biochemical diagnosis of ovarian carcinoma’ compares the expression of matrix metalloproteinases in ovarian cancer samples, samples from benign lesions, and healthy samples. It also compares the expression of these proteins with commonly used markers HE4 and CA125 in order to find additional markers that can be used for earlier diagnosis of ovarian cancer.
The manuscript provides good background information and the methods are described well. The experiments and analyses are fine however there are too many errors that should be corrected.
The following suggestions are provided:
1. There are too many grammatical and typographical errors that distracts the reader from the research findings. For example, the following statement in the simple summary should be changed ‘Collagenases (MMP-1, MMP-8 and MMP-13) are a poorly studied group of enzymes from the 28 MMPs, our study suggests that they may be postulated as new diagnostic markers of OC.’ to read ‘Collagenases (MMP-1, MMP-8 and MMP-13) are a poorly studied group of enzymes from the 28 MMPs. Our study suggests that they may be postulated as new diagnostic markers of OC.’ Similarly in line 117, it is mentioned ‘Patients in all groups were calculated ROMA…’. which is not grammatically correct. Please go through the whole manuscript to correct these errors.
2. In many places the healthy subjects’ samples (HS) are written as HL. Please be consistent on whether they should be referred to as HS or HL in the manuscript. For example, line 38, 168, etc.
3. In many places ‘begin’ is written instead of ‘benign’. For example, table 1, lines 194, 195, 205, etc. Please correct it everywhere.
4. Line 55 mentions that ‘In 2024 alone, 477,000 women will be diagnosed with OC…’. Since 2024 is almost coming to an end, this statement should be altered.
5. Please add the label/title to the y-axis for figure 6. It currently says % only.
6. The axes for figure 6 is not correct.
7. It is not clear what ‘otzyrmated’ refers to in line 294.
8. The lines 349-351 have erroneous information.
9. Since Zeng L, et al (line 480) also looked at MMP-13 in the context of ovarian cancer, line 369-370 should be modified.
10. Please show some figures of the correlations.
Comments on the Quality of English LanguageAs mentioned before, there are too many errors that must be corrected so that the article is well received by the reader.
Author Response
The manuscript titled ‘Diagnostic utility of metalloproteinases from collagenase group (MMP-1, MMP-8 and MMP-13) in biochemical diagnosis of ovarian carcinoma’ compares the expression of matrix metalloproteinases in ovarian cancer samples, samples from benign lesions, and healthy samples. It also compares the expression of these proteins with commonly used markers HE4 and CA125 in order to find additional markers that can be used for earlier diagnosis of ovarian cancer.
The manuscript provides good background information and the methods are described well. The experiments and analyses are fine however there are too many errors that should be corrected.
Dear Reviewer,
First of all, we would like to thank You sincerely for Your thorough and honest review of our article and for your time. Responses to the Reviewer's suggestions and questions will be written in green italics and any changes to the manuscript will be highlighted in green.
The following suggestions are provided:
- There are too many grammatical and typographical errors that distracts the reader from the research findings. For example, the following statement in the simple summary should be changed ‘Collagenases (MMP-1, MMP-8 and MMP-13) are a poorly studied group of enzymes from the 28 MMPs, our study suggests that they may be postulated as new diagnostic markers of OC.’ to read ‘Collagenases (MMP-1, MMP-8 and MMP-13) are a poorly studied group of enzymes from the 28 MMPs. Our study suggests that they may be postulated as new diagnostic markers of OC.’ Similarly in line 117, it is mentioned ‘Patients in all groups were calculated ROMA…’. which is not grammatically correct. Please go through the whole manuscript to correct these errors.
Thank You for Your comment and thoroughness during the review. The manuscript has been carefully reviewed again and the corrections suggested by the Reviewer have been made to the work - any changes are marked in green.
- In many places the healthy subjects’ samples (HS) are written as HL. Please be consistent on whether they should be referred to as HS or HL in the manuscript. For example, line 38, 168, etc.
Thank You for Your comment and we apologize very much for the inaccuracy. The manuscript has been corrected so that samples of healthy people are labeled as HS only. The changes have been highlighted in green.
- In many places ‘begin’ is written instead of ‘benign’. For example, table 1, lines 194, 195, 205, etc. Please correct it everywhere.
Thank You for Your comment and we are very apologetic for the occurrence of typos of this type. The manuscript has been rechecked and corrected. The changes have been highlighted in green.
- Line 55 mentions that ‘In 2024 alone, 477,000 women will be diagnosed with OC…’. Since 2024 is almost coming to an end, this statement should be altered.
Thank You for Your comment, the given sentence will be appropriately transformed in order not to mislead the potential reader - in the task we will emphasize that the figures are from October 2024. The change is marked in green color
- Please add the label/title to the y-axis for figure 6. It currently says % only.
Thank you for your comment The figure has been corrected accordingly.
- The axes for figure 6 is not correct
Thank You very much for Your comment and for your vigilance in reviewing our work. The description of Figure 6 has been corrected accordingly.
- It is not clear what ‘otzyrmated’ refers to in line 294.
Thank You for Your comment. In the given fragment of the work, a typo got in. We meant the word “obtained”. The change has been applied to the work (marked in green).
- The lines 349-351 have erroneous information.
Thank You for Your comment and we apologize for the error. The given passage has been corrected (the changes are marked in green).
- Since Zeng L, et al (line 480) also looked at MMP-13 in the context of ovarian cancer, line 369-370 should be modified.
Thank You for Your comment. However, the paper by Zeng et al. refers only to MMP-13 expression in ovarian cancer tissue. Our team, on the other hand, determined MMP-13 concentrations in peripheral blood. However, we realize that the section of the manuscript highlighted by the reviewer may be inaccurate, so it has been modified accordingly. The changes have been highlighted in green.
- Please show some figures of the correlations.
Thank You for your comment; however, we feel that the figures showing the correlations between the compounds studied are unnecessary. We performed correlation determinations using Spearman's method and did not show statistically significant correlations - therefore, the determined correlations are included in the Supplementary File. Therefore, we have not added correlation figures to the manuscript, and with all due respect to the reviewer, but we believe that such figures are unnecessary.
Comments on the Quality of English Language - As mentioned before, there are too many errors that must be corrected so that the article is well received by the reader.
Thank You for Your comment and again we apologize for the large number of grammatical errors and typos. The manuscript has been carefully reviewed again. In addition, we have corrected all the errors that the Reviewer found. We hope that the manuscript in its current form will be accepted.
In conclusion, we sincerely thank You for such a positive evaluation of our manuscript. We hope that the answers we presented to the Reviewer's questions and the changes we made to the paper will be satisfactory and our manuscript will be accepted for publication in the “Cancers”.
Best regards,
Aleksandra Kicman
prof. dr hab. Sławomir Ławicki
also, on behalf of all authors

Reviewer 3 Report
Comments and Suggestions for Authors
In this study the authors aimed to determine the diagnostic utility of collagenases (MMP-1, MMP-8 and MMP-13) in the diagnosis of Ovarian carcinoma (OC) compared to the currently diagnostic markers of OC (HE4, CA125) and the Risk of Ovarian Malignancy Algorithm (ROMA). Overall, this is a very interesting study that contributes with promising preliminary data for potentially expanding OC diagnostic capabilities, which is especially important given the asymptomatic progression and poor prognosis associated with OC.
The study is well executed, and the authors highlight in the discussion the major limitations of the study which include a small sample size, reliance on only two analytical methods to quantify MMPs (ELISA and CMIA), and the lack of tissue expression analysis of the collagenases.
Specific comments:
Introduction
Line 67: The sentence “The issues of OC diagnosis, are not only related to the mostly asymptomatic course of the disease and the lack of effective screening tests.” in not clear. Consider revising.
Line 78: The term 'modern tumor markers' is quite vague and lacks a precise definition.
Consider expanding the reasoning for choosing to study the collagenase group metalloproteinases - MMP-1, MMP-8 and MMP-13- in the introduction. This is clearly stated only in the discussion.
Materials and Methods
Line 102-103: It is not clear what is the meaning of “by introducing the treatment” in the sentence.
Line 124-125: It is not clear what is the meaning of “treated with oncology.” in the sentence.
All the healthy women (HS group) in the study were post-menopausal. Couldn’t this introduce a bias, since in the other groups there are both pre- and post-menopausal women? This is not discussed.
Figure 1. Consider including "Collection of blood" before the centrifugation step in the Flowchart. Consider revising the title. For example, “Flowchart of experimental design”.
Results
Line 250-253: Results presented refer to NPV and not PPV.
Line 267: Should be Figures 6-8.
Line 274: Should be Figure 6
Line 294: What does it mean “otzrymated” in the sentence?
Line 300: Should be Table 4.
Author Response
In this study the authors aimed to determine the diagnostic utility of collagenases (MMP-1, MMP-8 and MMP-13) in the diagnosis of Ovarian carcinoma (OC) compared to the currently diagnostic markers of OC (HE4, CA125) and the Risk of Ovarian Malignancy Algorithm (ROMA). Overall, this is a very interesting study that contributes with promising preliminary data for potentially expanding OC diagnostic capabilities, which is especially important given the asymptomatic progression and poor prognosis associated with OC.
The study is well executed, and the authors highlight in the discussion the major limitations of the study which include a small sample size, reliance on only two analytical methods to quantify MMPs (ELISA and CMIA), and the lack of tissue expression analysis of the collagenases.
Dear Reviewer,
First of all, we would like to thank You sincerely for Your thorough and honest review of our article and for Your time. Responses to the Reviewer's suggestions and questions will be written in green italics and any changes to the manuscript will be highlighted in green.
Specific comments:
Introduction
Line 67: The sentence “The issues of OC diagnosis, are not only related to the mostly asymptomatic course of the disease and the lack of effective screening tests.” in not clear. Consider revising.
Thank You for Your comment. Ovarian cancer in the vast majority of cases is asymptomatic or sparse - in this case, the woman usually confuses these symptoms with complaints from other systems. As for screening, currently medicine has only transvaginal ultrasonography and determination of tumor markers (HE4 and CA125) however, these methods are of low sensitivity. We understand that the sentence from the introduction is little understood. According to the Reviewer's suggestion, the given part of the introduction will be transformed. The change is marked in green.
Line 78: The term 'modern tumor markers' is quite vague and lacks a precise definition.
Thank You for Your comment, the term “modern tumor markers” is indeed unclear. As suggested by the Reviewer, they will be referred to as “tumor markers”. The given definition has been changed and highlighted in green.
Consider expanding the reasoning for choosing to study the collagenase group metalloproteinases - MMP-1, MMP-8 and MMP-13- in the introduction. This is clearly stated only in the discussion.
Again, thank You for Your comment. As suggested by the reviewer, a brief rationale for the selection of collagenases for the study will be added to the introduction. The change has been highlighted in green.
Materials and Methods
Line 102-103: It is not clear what is the meaning of “by introducing the treatment” in the sentence.
Thank You for Your comment. Written consent was taken from all subjects who participated in the study. This consent was taken prior to the introduction of treatment - usually the same one in which the patient was admitted to the hospital. The phrase “introduction of treatment” refers to any medical treatment the patients underwent. However, at the request of the Reviewer, the given sentence was modified accordingly. The change has been highlighted in green.
Line 124-125: It is not clear what is the meaning of “treated with oncology.” in the sentence.
Thank You for Your comment. The given phrase refers to any oncology treatment. Patients previously treated with oncology did not have the ROMA algorithm calculated. The given phase was transformed accordingly.
All the healthy women (HS group) in the study were post-menopausal. Couldn’t this introduce a bias, since in the other groups there are both pre- and post-menopausal women? This is not discussed.
Thank You for Your comment. The control group included only postmenopausal patients - this is due to the continuation of our previous studies (https://doi.org/10.3390/ijms25116265). In order to maintain continuity of experience, in that study as well, we decided to select only postmenopausal patients for the control group. The selection of such a control group was also dictated by the initial large disparity in the number of premenopausal and postmenopausal women. At the stage of sample collection, the vast majority of the material came from postmenopausal women, so in order not to introduce such a large disproportion, we decided to determine only in the group of postmenopausal women. We hope for understanding from the Reviewer.
Figure 1. Consider including "Collection of blood" before the centrifugation step in the Flowchart. Consider revising the title. For example, “Flowchart of experimental design”.
Thank You for Your rightful comment, the figure and its caption have been corrected as suggested by the reviewer. The changes have been highlighted in green.
Results
Line 250-253: Results presented refer to NPV and not PPV.
Thank You for Your comment and we apologize for the mistake. The change has been applied to the manuscript and highlighted in green.
Line 267: Should be Figures 6-8.
Thank You for Your comment and we apologize for the mistake. The change has been applied to the manuscript and highlighted in green.
Line 274: Should be Figure 6
Thank You for Your comment and we apologize for the mistake. The change has been applied to the manuscript and highlighted in green.
Line 294: What does it mean “otzrymated” in the sentence?
Thank You for Your comment. In the given fragment of the work, a typo got in. We meant the word “obtained”. The change has been applied to the work (marked in green).
Line 300: Should be Table 4.
Thank You for Your comment and we apologize for the mistake. The change has been applied to the manuscript and highlighted in green.
In conclusion, we sincerely thank You for such a positive evaluation of our manuscript. We hope that the answers we presented to the Reviewer's questions and the changes we made to the paper will be satisfactory and our manuscript will be accepted for publication in the “Cancers”.
Best regards,
Aleksandra Kicman
prof. dr hab. Sławomir Ławicki
also, on behalf of all authors

Round 2
Reviewer 1 Report
Comments and Suggestions for Authors
The author evaluated the diagnostic utility of collagenases (MMP-1, MMP-8 and MMP-13) in the diagnosis of epithelial ovarian cancer (EOC) compared to HE4 and CA125 and 34 the ROMA. The authors concluded that MMP-1 and MMP-13 have shown preliminary potential as diagnostic markers and auxiliary markers to ROMA in biochemical diagnosis of EOC. This research cannot be used immediately in clinical practice. However, I think it will be interesting for cancers readers. I wrote some specific comments for reviewing the original manuscript. The authors answered clearly in Response to Decision Letter and revised adequately. I think that their revision and response meet my requirement.